# Outcomes and Loop Pattern Analysis of a Road-Map Technique for ERCP with Side-Viewing Duodenoscope in Patients with Billroth II Gastrectomy (with Video)

**DOI:** 10.3390/jpm11050404

**Published:** 2021-05-12

**Authors:** Min Jae Yang, Jin Hong Kim, Jae Chul Hwang, Byung Moo Yoo, Yu Ji Li, Soon Sun Kim, Sun Gyo Lim

**Affiliations:** Department of Gastroenterology, Ajou University School of Medicine, 164 Worldcup-ro, Yeongtong-gu, Suwon 16499, Korea; creator1999@hanmail.net (M.J.Y.); cath07@naver.com (J.C.H.); ybm6403@gmail.com (B.M.Y.); mikou2000@163.com (Y.J.L.); cocorico99@gmail.com (S.S.K.); mdlsk75@gmail.com (S.G.L.)

**Keywords:** cholangiopancreatography endoscopic retrograde, gastrectomy, duodenoscopy, catheters, pneumoperitoneum

## Abstract

Endoscopic retrograde cholangiopancreatography (ERCP) in patients who have undergone a Billroth II gastrectomy is a major challenge. This study aimed to evaluate the outcomes of the road-map technique for duodenal intubation using a side-viewing duodenoscope for ERCP in Billroth II gastrectomy patients with naïve papilla, and to analyze the formation and release patterns of common bowel loops that occur when the duodenoscope navigates the afferent limb. The duodenoscopy approach success rate was 85.8% (97/113). In successful duodenoscopy approach patients, there were five bowel looping patterns that occurred when the preceding catheter-connected duodenoscope was advanced into the duodenum: (1) reverse ɣ-loop (29.9%), (2) fixed reverse ɣ-loop (5.2%), (3) simple U-loop (22.7%), (4) N-loop (28.9%), and (5) reverse alpha loop (13.4%). The duodenoscopy cannulation and duodenoscopy therapeutic success rates were 81.4% (92/113) and 80.5% (91/113), respectively, while the overall cannulation and therapeutic success rates were 92.0% (104/113) and 87.6% (99/113), respectively. Bowel perforation occurred in three patients (2.7%). The road-map technique may benefit duodenoscope-based ERCP in Billroth II gastrectomy patients by minimizing the tangential axis alignment between the duodenoscopic tip and driving of the afferent limb, and by predicting and counteracting bowel loops that occur when the duodenoscope navigates the afferent limb.

## 1. Introduction

Endoscopic retrograde cholangiopancreatography (ERCP) is an established procedure for the diagnosis and treatment of pancreatobiliary diseases [1]. Unlike in patients with normal anatomy, there are multiple hurdles to successfully complete therapeutic ERCP in Billroth II gastrectomy patients. These include difficulties surrounding the entrance to the afferent limb, reaching and visualizing the papilla, cannulating the papilla with a reverse orientation, and performing therapeutic maneuvers [2]. Various endoscope types, including a side-viewing duodenoscope, forward-viewing endoscope, and balloon enteroscope, have been used for ERCP in patients with Billroth II anatomy, depending on local experience and the reconstruction anatomy type [3]. While forward-viewing endoscopes provide safe intubation and navigation of the afferent limb under direct visualization, biliary cannulation and therapeutic maneuvers are often challenging owing to the tangential view of the papilla and the limited use of dedicated ERCP devices [3,4]. Conversely, side-viewing duodenoscopes facilitate cannulation and therapeutic maneuvers owing to the en face view of the papilla, elevator presence, and large working channel [2]. However, the major drawbacks of this duodenoscope are the difficulty to see the bowel lumen en face and limited bend of the distal tip of the endoscope with a large turning radius, which leads to an increased perforation risk when passing through the angulated afferent loop [5,6].

A road-map technique using an anticipative guidewire and catheter has been instrumental for fluoroscopic and endoscopic guidance in advance of duodenoscopic progression in the afferent loop [2,7,8]. The aims of this study were to evaluate the outcomes of the road-map technique for duodenal intubation using a side-viewing duodenoscope for ERCP in Billroth II gastrectomy patients and to analyze the formation and release patterns of common bowel loops that occur when the duodenoscope navigates the afferent limb.

## 2. Materials and Methods

### 2.1. Patients

Between January 2003 and December 2019, patients with a previous Billroth II gastrectomy who underwent ERCP and fulfilled the eligibility criteria were retrospectively reviewed from a prospectively designed ERCP database system at the Ajou University Hospital (Figure 1). The eligibility criteria were: (1) prior Billroth II gastrectomy with or without Braun anastomosis, (2) naïve papilla, (3) ERCP initiated with a side-viewing duodenoscope, and (4) road-map technique using a preceding guidewire and catheter technique used for afferent limb navigation. We excluded patients with prior ERCP history, patients in whom ERCP was initiated with a forward-viewing endoscope, and patients with peritoneal carcinomatosis.

### 2.2. ERCP Procedures

ERCP was initiated with the patient in the prone position using a side-viewing duodenoscope (JF-260V or TJF-260V; Olympus Corp., Tokyo, Japan) under conscious sedation with standard doses of midazolam, propofol, and meperidine. The road-map technique was performed as follows (Appendix A): (1) after successful intubation of the targeted jejunal limb, an angle-tip guidewire preloaded catheter, such as a triple-lumen balloon catheter (Quattro; Cook Endoscopy Inc, Limerick, Ireland) or rotatable sphincterotome (TRUEtome; Boston Scientific, Natick, MA, USA) was introduced through the working channel of the duodenoscope, and the guidewire and catheter were sequentially negotiated into the intubated jejunal limb. Successful afferent limb intubation was determined comprehensively considering the presence of bile, enterographic findings, and the driving course of the guidewire and catheter. (2) If the preceding guidewire and catheter successfully reached the duodenal stump, the duodenoscope was cautiously advanced toward the papilla over the catheter, referencing the preceding catheter under fluoroscopy and endoscopy. (3) During duodenoscope advancement, the direction of the preceding catheter was repeatedly checked using an elevator under endoscopy to predict the luminal direction of the bowel. In accordance with duodenoscope advancement, the preceding catheter was repeatedly pulled back under fluoroscopic guidance to ensure the shortest distance to the duodenal stump and to prevent loop formation near the ligament of Treitz.

In cases with successful duodenoscopic access to the papilla, selective cannulation was primarily attempted using a straight cannula (ERCP catheter; MTW Endoscopie, Wesel, Germany), bendable-tip cannula (SwingTip; Olympus Corp, Tokyo, Japan), or rotatable sphincterotome (TRUEtome; Boston Scientific, Natick, MA, USA). Endoscopic sphincterotomy was performed using a rotatable sphincterotome (TRUEtome; Boston Scientific, Natick, MA, USA), Billroth II sphincterotome (PTG-20-6-BII-NG; Cook Endoscopy Inc, Limerick, Ireland), or needle knife (Micro-knife, Boston Scientific, Natick, MA, USA). Other therapeutic interventions were performed using standard ERCP accessories. In cases of failed duodenoscopic access to the papilla, the duodenoscope was changed to a cap-pitted forward-viewing double-channel gastroscope or colonoscope. In all patients, carbon dioxide was supplied during ERCP.

### 2.3. Outcome Measurements

The outcomes of ERCP, including success of the approach, cannulation, intended interventions, and adverse events, were evaluated. Furthermore, the formation and release patterns of common looping that occurred when the duodenoscope navigated the afferent limb were analyzed (Figure 2 and Figure 3; Appendix A). Duodenoscopy approach success was defined as success in reaching the duodenal stump and visualizing the major papilla with a duodenoscope. The bowel loop pattern was analyzed in patients who achieved duodenoscopy approach success, because it may be inaccurate in situations where the duodenoscope could not be advanced beyond the ligament of Treitz. Duodenoscopy cannulation success was defined as successful guidewire placement deep in the targeted biliopancreatic duct using a duodenoscope. Duodenoscopy therapeutic success was defined as duodenoscopy approach and cannulation success, with achievement of the planned therapeutic goals using a duodenoscope. Overall success was defined as the achievement of the intended procedure during admission for the same, regardless of the number of ERCP sessions, using a duodenoscope, rescue forward-viewing endoscope, or percutaneous-endoscopic rendezvous technique. Adverse events were defined and graded according to the American Society for Gastrointestinal Endoscopy severity grading system [9] and Tokyo 18 guidelines [10,11].

Appendix A: a road-map technique for duodenal intubation of a side-viewing duodenoscope and five characteristic loop patterns of advancing a duodenoscope in the afferent loop.

### 2.4. Statistical Analyses

Mean and standard deviation (SD) were used to describe continuous variables, while categorical variables were described as percentages and analyzed using the chi-square test. Statistical analyses were performed using IBM SPSS statistics software (version 25.0; IBM, Armonk, NY, USA).

## 3. Results

This study enrolled a total of 113 patients who underwent a Billroth II gastrectomy, including 29 patients with Braun anastomosis. A flow diagram of the study cohort is shown in Figure 1. The baseline characteristics of the patients are presented in Table 1. The reasons for ERCP were suspicious or radiologically confirmed common bile duct stones in 83 patients (73.5%), indeterminate biliary stricture in six patients (5.3%), and malignant biliary stricture by pancreatobiliary malignancy or recurrent gastric cancer in 21 patients (18.6%).

ERCP outcomes are presented in Table 2 and Figure 1. A road-map technique was performed in 73 patients with a rotatable sphincterotome (64.6%) and in 40 patients with a triple-lumen balloon catheter (35.4%). The duodenoscopy approach success rate was 85.8% (97/113) (88.1% without Braun anastomosis, 79.3% with Braun anastomosis, *p* = 0.242). The reasons for duodenoscopy approach failure included bowel perforation in three patients, long U-loop formation due to bowel redundancy before reaching the ligament of Treitz in nine patients, and failure to pass the angulated ligament of Treitz in four patients. Among the latter 13 patients, seven patients had successful papillary access, which was achieved using a rescue cap-pitted forward-viewing endoscope.

In patients with duodenoscopy approach success (*N* = 97), there were five characteristic patterns of bowel looping that occurred when the preceding catheter-connected duodenoscope was advanced into the duodenum beyond the ligament of Treitz (Table 3; Figure 2 and Figure 3; Appendix A): (1) reverse ɣ-loop (29.9%), (2) fixed reverse ɣ-loop (5.2%), (3) simple U-loop (22.7%), (4) N-loop (28.9%), and (5) reverse alpha loop (13.4%).

Duodenoscopy cannulation and duodenoscopy therapeutic success rates were 81.4% (92/113) and 80.5% (91/113), respectively, while the overall cannulation and therapeutic success rates were 92.0% (104/113) and 87.6% (99/113), respectively. Among 22 patients with duodenoscopy therapeutic failure, overall therapeutic success was finally achieved with rescue cap-pitted forward-viewing endoscope in four patients and percutaneous-endoscopic rendezvous technique in four patients. The rescue methods for overall therapeutic failure are presented in Table 2 and Figure 1.

ERCP-related adverse events occurred in 8/113 patients (7.1%) (Table 2). Duodenoscope-related bowel perforation occurred in three patients. In two patients, perforation occurred because of a misaligned axis between the duodenoscope and the jejunum when the duodenoscope was pushed from the inferior angle of the proximal jejunum upward toward the ligament of Treitz, thereby creating a large U-loop. In another patient, perforation occurred while turning the duodenoscope tip to solve a reverse gamma loop, which had formed near the ligament of Treitz. All perforations occurred within the first five years, and no perforation occurred during the subsequent years. No procedure-related mortality was observed.

## 4. Discussion

The endoscope choice for ERCP in Billroth II gastrectomy patients is controversial. The prerequisites of an ideal endoscope are to guarantee safe endoscopic access through the angulated afferent limb and to embody the en face view of the papilla for selective cannulation and therapeutic interventions. Although many experienced endoscopists advocate the routine use of a side-viewing duodenoscope for ERCP in patients with Billroth II anatomy [2,7], previous studies have reported that the incidence of bowel perforation is higher with a side-viewing duodenoscope than with a forward-viewing endoscope [5,12]. Moreover, in a recent meta-analysis, procedure-related mortality occurred exclusively in the side-viewing duodenoscopy group [12]. To guide safe afferent limb navigation, anticipative insertion of a guidewire and catheter into the duodenal stump before duodenoscope advancement was attempted and showed a therapeutic success rate of 81.3–86.3%, a perforation rate of 0.0–1.8%, and procedure-related mortality of 0.0–0.3% in Billroth II gastrectomy patients [2,7,8]. The preceding catheter can act as a fluoroscopic and endoscopic road map for subsequent duodenoscope advancement, either with the catheter itself or by providing enterography. Furthermore, the approach time can be shortened by navigating the three bowel entrances at the jejunojejunal anastomosis in Braun anastomosis with a catheter preceding the duodenoscope.

Of the various catheters available, we used a triple-lumen balloon catheter and rotating sphincterotome in this study. The triple-lumen balloon catheter tip (Quattro; Cook Medical, USA) is buried by the balloon when it is maximally inflated. Therefore, the fully inflated balloon catheter can be pushed into and smoothly explore the afferent loop without bowel injury. Sometimes, the fully inflated balloon catheter itself can pass through the angulated portion of the small bowel more easily than the guidewire. Moreover, it can provide better imaged enterography by balloon occlusion. The rotating sphincterotome has the advantage of guidewire negotiation near the ligament of Treitz by changing the catheter tip direction.

In the current study, bowel perforation occurred in one patient in the ascending jejunal segment before reaching the ligament of Treitz, just after passing the inferior angle of the proximal jejunum. In this intestinal region, the duodenoscope is usually pushed out, creating a large U-loop, while the duodenoscope tip is bent in a limited manner, leading to a larger turning radius than that observed with a forward-viewing gastroscope. Therefore, the axis between the duodenoscope and jejunal lumen is likely to be misaligned when pushing into the ascending jejunal segment and progressing to the duodenojejunal junction. To adjust the axis and avoid perforation, the preceding catheter direction should be repeatedly checked under endoscopy using a duodenoscope elevator for luminal direction prediction of the afferent limb (Appendix A). In addition, the preceding catheter should be repeatedly pulled back under fluoroscopic guidance in correspondence with duodenoscope advancement to reduce the resistance against the duodenoscope and prevent loop formation near the ligament of Treitz. This will also ensure the shortest distance to the duodenal stump.

As shown in the current study, the duodenojejunal junction, which is relatively fixed by the ligament of Treitz, is another dangerous site with a high perforation risk. Therefore, it is important to be aware of common loop patterns that are likely to be formed near the ligament of Treitz. In reviewing our data, we found that there are five characteristic patterns of bowel looping that occur when the duodenoscope advances beyond the ligament of Treitz in the afferent limb in Billroth II gastrectomy patients (Figure 2 and Figure 3; Appendix A).

Reverse gamma loop formation near the ligament of Treitz is the most common bowel loop pattern. In this situation, the duodenoscope cannot be propelled beyond the duodenojejunal junction using the pushing method because of the tangential axis alignment between the duodenoscopic tip and the driving of the duodenum. Hence, the loop should be corrected by hooking the duodenoscopic tip into the duodenojejunal junction, pulling it back, rotating it toward the right, and making the distal portion of the duodenoscope similar to a turning duck’s head (Figure 2a). However, in some cases, the reverse gamma loop cannot be released because of bowel fixation by the ligament of Treitz. In these patients, papillary access can occasionally be achieved with the pushing method if the axis between the duodenoscopic tip and the driving of the duodenum is not misaligned (Figure 2b). This also creates a larger reverse gamma loop.

When only a simple U-loop is formed in the afferent limb, the road-map technique has a limited role because the duodenoscope can easily approach the papilla (Figure 3a). In the formation of an N-loop, the duodenoscope approach is more difficult than in a simple-U loop. As shown in Figure 3b, in this procedure, the duodenoscopic view faces cranially, with the preceding catheter running over the duodenoscopic tip toward the back side of the duodenoscope, similar to a backstroke. This is the second most common bowel loop formed in the afferent limb.

With a reverse alpha loop formation in the afferent loop (Figure 3c), the duodenoscopic view faces caudally and can be propelled without axis misalignment if the preceding catheter has already successfully reached the duodenal stump. After intubating the duodenal second portion with the duodenoscope, the loop can be easily corrected by pulling back the duodenoscope and rotating it left.

In our experience, the reverse gamma loop, N-loop, and reverse alpha loop patterns can be converted into each other (Appendix A) if the intraabdominal adhesion is not severe and the afferent limb is not firmly fixed by the ligament of Treitz. As the reverse gamma loop is more difficult to release than other loops, it would be better to convert it to the N-loop or reverse alpha loop. If the preceding guidewire is successfully advanced into the duodenal stump but its driving is formed in the reverse gamma loop pattern, the loop of the preceding guidewire should be converted to an N- or reverse alpha loop by pulling back and rotating the guidewire before duodenoscope advancement beyond the inferior angle of the proximal jejunum. Next, the preceding catheter should be advanced over the guidewire, up to the duodenal stump, without gamma loop reformation. Finally, the duodenoscope should be advanced over the preceding catheter while pulling the catheter back in correspondence with the forward movement of the duodenoscope to prevent gamma loop reformation.

Recently, short-type single and double balloon enteroscope-assisted ERCP has gained popularity as the procedure of choice for pancreatobiliary disease treatment in Billroth II gastrectomy patients. This technique has shown an outstanding therapeutic success rate of nearly 100%, with a negligible perforation rate [13,14]. These scopes have a 3.2 mm-large working channel and a 152 cm working length, which allows the use of most conventional ERCP accessories. Moreover, these scopes show improved bending capacity and a short turn radius in the distal endoscopic tip, which enables an en face view of the papilla despite being forward-viewing in nature. Therefore, future well-designed prospective studies comparing balloon enteroscope-assisted ERCP and road-map technique-assisted duodenoscope-based ERCP are needed to accumulate reliable data for the determination of an ideal endoscope for Billroth II gastrectomy patients.

The strength of the current study is that it classified the bowel loop patterns of the afferent limb for the first time. Using this analysis, we then suggested technical additions to the road-map technique to overcome the uncertainty of afferent limb navigation with a side-viewing duodenoscope in patients with Billroth II anatomy. However, our study had several limitations. The main limitation was the potential bias related to the retrospective single-arm design of our study in a single center. Although we used a prospectively designed organized fill-in ERCP database to record the demographics and ERCP data of each patient for comprehensive data collection, there was still insufficient evidence to clarify the usefulness of the road-map technique as the standard choice for ERCP in Billroth II gastrectomy patients. Furthermore, there may be additional bowel loop patterns that were not identified in our study. Finally, in the current study, we encountered three bowel perforation cases during the duodenoscope approach in the afferent limb, despite the road-map technique. However, a learning curve for ERCP in Billroth II patients was present, and no perforation occurred after the first five years.

In conclusion, the road-map technique can benefit duodenoscope-based ERCP in Billroth II gastrectomy patients by minimizing the tangential axis alignment between the duodenoscopic tip and the driving of the afferent limb, and by predicting and counteracting the common bowel loops that occur when the duodenoscope navigates the afferent limb. Further data accumulation through future well-designed large-scale studies is warranted to determine the ideal endoscope and approach technique tailored to each Billroth II gastrectomy patient.

## Figures and Tables

**Figure 1 jpm-11-00404-f001:**
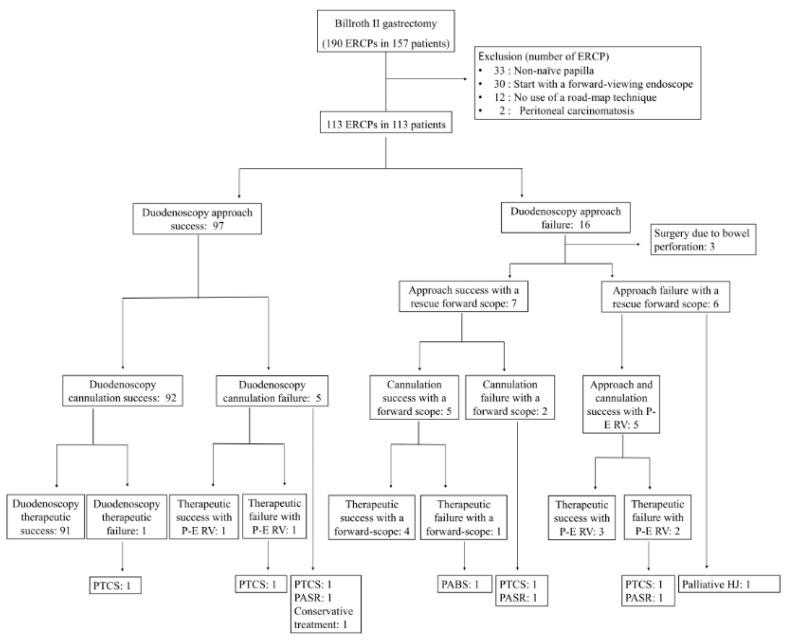
A flow diagram for study cohort. P-E RV, percutaneous-endoscopic rendezvous; PTCS, percutaneous transhepatic cholangioscopy; PASR, percutaneous antegrade stone removal; PABS, percutaneous antegrade biliary stenting.

**Figure 2 jpm-11-00404-f002:**
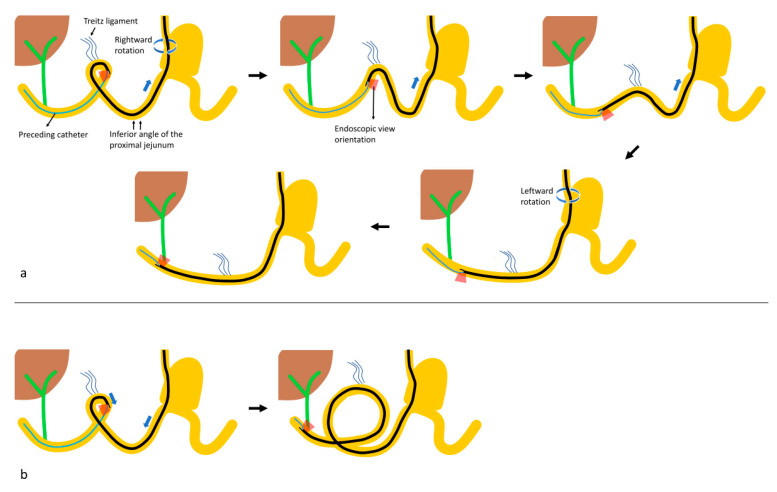
The loop patterns of inserting a duodenoscope in the afferent loop. (**a**) Reverse gamma (ɣ) loop, (**b**) fixed reverse gamma loop.

**Figure 3 jpm-11-00404-f003:**
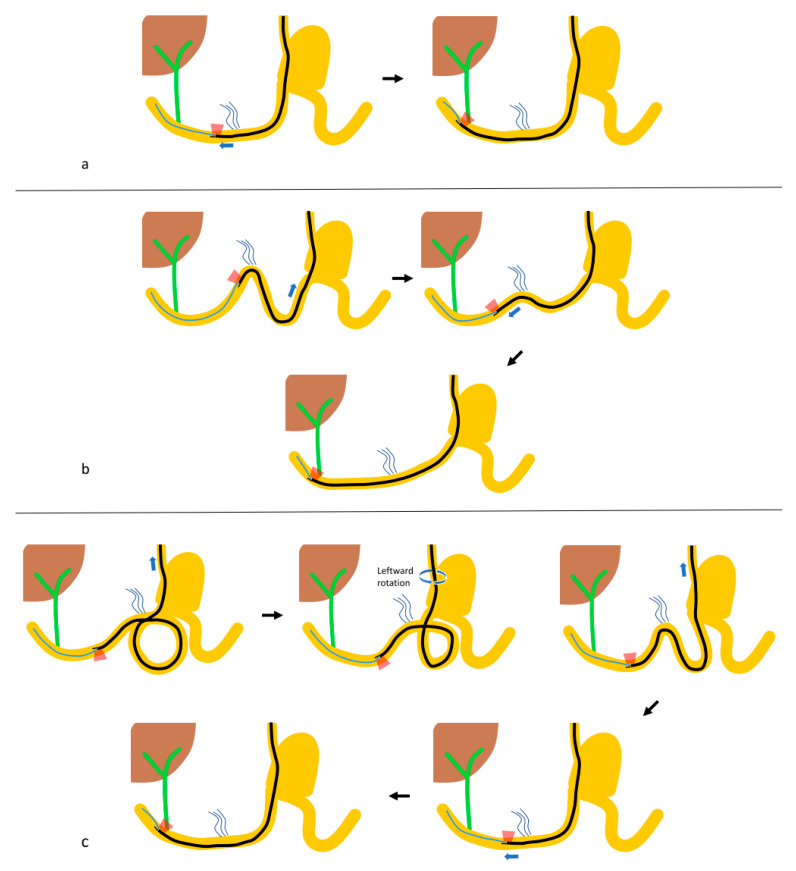
The loop patterns of inserting a duodenoscope in the afferent loop. (**a**) Simple U-loop, (**b**) N-loop, (**c**) reverse alpha (α) loop.

**Table 1 jpm-11-00404-t001:** Baseline characteristics of the patients (*N* = 113).

Age, mean ± SD	70.9 ± 9.4
Sex, n (%)	84 (74.3)/29 (25.7)
Reconstruction type, n (%)	
Billroth II/Billroth II with Braun anastomosis	84 (74.3)/29 (25.7)
Antecolic/retrocolic	83 (73.5)/30 (26.5)
Billroth II gastrectomy indication, n (%)	
Gastric cancer/peptic ulcer	78 (69.0))/35 (31.0)
Duration from Billroth II gastrectomy, n (%)	
≤5 year/>5 years	25 (22.1)/88 (77.9)
Reasons for ERCP, n (%)	
Common bile duct stone suspicion	83 (73.5)
Indeterminate biliary stricture	6 (5.3)
Malignant biliary stricture	21 (18.6)
Recurred gastric cancer	6
Cholangiocarcinoma (hilar/distal)	8 (2/6)
Pancreatic cancer	4
Ampullary cancer	2
Gallbladder cancer	1
Bile leakage	1 (0.9)
Pancreatic duct leakage/chronic pancreatitis	2 (1.8)
Periampullary diverticulum, n (%)	33 (29.7)
Preprocedural laboratory finding, mean ± SD	
White blood cell,/µL	8076 ± 4155
Hemoglobin, g/dL	11.8 ± 1.6
Total bilirubin, mg/dL	3.2 ± 4.2

SD, standard deviation; ERCP, endoscopic retrograde cholangiopancreatography.

**Table 2 jpm-11-00404-t002:** Endoscopic retrograde cholangiopancreatography outcomes in Billroth II gastrectomy patients (*N* = 113).

Used catheter, n (%)	
Rotatable sphincterotome/triple-lumen balloon catheter	73 (64.6)/40 (35.4)
Duodenoscopy/overall approach success, n (%)	97 (85.8)/109 (96.5)
Causes of duodenoscopy approach failure, n	16
Bowel perforation occurrence	2
Long U-loop formation before reaching the ligament of Treitz	10
Failure in passing the ligament of Treitz	4
Duodenoscopy/overall cannulation success, n (%)	92 (81.4)/104 (92.0)
Duodenoscopy/overall therapeutic success, n (%)	91 (80.5)/99 (87.6)
Reasons for overall therapeutic failure, n	14
Approach failure albeit forward scope or P-E RV	1
Cannulation failure/occurrence of bowel perforation	6/3
Failure in complete stone extraction	3
Failure in biliary stenting due to tight stricture	1
Rescue methods for overall therapeutic failure, n (%)	
PTCS/percutaneous antegrade stone removal	5/3
Percutaneous antegrade stenting/palliative hepaticojejunostomy	1/1
Conservative treatment	1
Surgery following bowel perforation	3
Primary repair and cholecystectomy	1
Primary repair and cholecystectomy and choledochotomy	2
Overall adverse events, n (%)	8 (7.1)
Duodenoscope-related perforation	3(2.7)
Pancreatitis	3 (2.7)
Cholecystitis	1 (0.9)
Transient respiratory failure	1 (0.9)

P-E RV, percutaneous-endoscopic rendezvous; PTCS, percutaneous transhepatic cholangioscopy.

**Table 3 jpm-11-00404-t003:** Loop patterns analysis of inserting a duodenoscope in the afferent loop in patients with duodenoscopy approach success (*N* = 97).

Reverse ɣ-loop *	29 (29.9%)
Fixed reverse ɣ-loop	5 (5.2%)
Simple U-loop	22 (22.7%)
N-loop *	28 (28.9%)
Reverse α-loop *	13 (13.4%)

* Three loop patterns could be converted into each other if the intraabdominal adhesion was not severe and the afferent limb was not firmly fixed by the ligament of Treitz.

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
