# Peer review of "Outcomes and Loop Pattern Analysis of a Road-Map Technique for ERCP with Side-Viewing Duodenoscope in Patients with Billroth II Gastrectomy (with Video)"

_jpm, 2021, doi:10.3390/jpm11050404_

Round 1

Reviewer 1 Report

The manuscript entitled "Outcomes and loop pattern analysis of a road-map technique 2 for ERCP with side-viewing duodenoscope in patients with 3 Billroth II gastrectomy" by Min Jae Yang et al. presents interesting and impressive data about the challenge of ERCP in patients after BII gastrectomy.

The manuscript is surely of big interest for gastroenterologists and surgeons and can be generally recommended for publication in JPM. However, there are some minor points which should be addressed before:

  • It would be helpful if the authors could present a figure or graph that shows clearly successful and non-successful interventions with and without road-map technique. Is the rate of success statistically higher with the road-map technique?
  •   What have been the reason for failure? Could the authors evaluate the loop patterns in these patients? Are there any negative predictive factors? 

Author Response

We would like to thank you for taking the time to review our manuscript and for your thoughtful suggestions and insights. We believe that our manuscript has benefited from these insightful suggestions. The detailed comments were all valid. We have carefully considered all reviewer comments, and revised our manuscript as suggested.

Responses to the comments of Reviewer 1

Comment #1: It would be helpful if the authors could present a figure or graph that shows clearly successful and non-successful interventions with and without the road-map technique. Is the rate of success statistically higher with the roadmap technique?

Response #1: We agree with your comment. In the current study, we did not show the objective data that the road-map technique provides advantages over standard techniques. However, during the study period, the road-map technique was the procedure of choice for ERCP in Billroth II gastrectomy patients in our institution, and it was implemented in most cases, except for a few cases with simple U-loops. Hence, we could not present comparative data regarding the superiority of the road-map technique over the standard techniques in this study. The lack of a control group was described as the main limitation of our study in the Discussion section. We will consider the reviewer’s insightful comment in our future prospective research.

Comment #2: What have been the reason for failure? Could the authors evaluate the loop patterns in these patients? Are there any negative predictive factors? 

Response #2-1: In the current study, there were 16 cases with duodenoscopy approach failure, albeit with the road-map technique. The reasons for duodenoscopy approach failure are listed in Table 2. The main reason for this was the long U-loop formation before reaching the ligament of Treitz. This was a situation that could not be solved with a duodenoscope with the road-map technique itself because it is essentially related to the Billroth II anatomy with a long limb. Other reasons were associated with limited visibility of the duodenoscope or the occurrence of unintended bowel perforation. This has been described in the Results section. Despite the use of the road-map technique, bowel perforations were unavoidable in some cases because the duodenoscope tip was bent in a limited manner, leading to a larger turning radius than that observed with a standard gastroscope. Therefore, the axis between the duodenoscope and jejunal lumen was likely to be misaligned in the angulated bowel segment despite the road-map technique. This issue has also been described in the Discussion section.

Response #2-2: In cases with duodenoscopy approach failure, we did not analyze the bowel loop pattern. This was because the bowel loop patterns could be interchangeable according to duodenoscope manipulation, as described in the supplementary video. Accordingly, bowel loop pattern analysis could be inaccurate in situations where the duodenoscope could not be advanced beyond the ligament of Treitz. This information was newly added to the outcome measurement section as follows: “Bowel loop pattern was analyzed in patients who achieved duodenoscopy approach success, because it may be inaccurate in the situation where the duodenoscope could not be advanced beyond the ligament of Treitz.”

Response #2-3: In our experience, postoperative adhesion near the ligament of Treitz, long afferent limb, reverse gamma loop formation near the ligament of Treitz, and peritoneal carcinomatosis related to recurrent tumors may be negative predictors for the failure of duodenoscopy approach. However, we did not verify these potential predictors through objective multivariate analysis because of the lack of information related to its retrospective nature. We plan to conduct an additional prospective study to clarify this issue.

Reviewer 2 Report

This retrospective study is a well conducted Analysis of a smart technique to perform ERCP on patients with a BillrothII resection. It is well written and undertsandable. Especially the Video illustrates the patterns well. As noted in the limitations it is a non controlled retrospective Observation. Ideally a control group would give an impression on the relative performance.

Why were patients older than 20 years eligible? How many under 20 received ERCP?

Generally the authors should check for spelling and grammar errors. Also in the provided document the formatization of the document is not well done (different font sizes etc.). I do wonder why this very clinical work has been submitted to the Journal of Personalized Medicine. What is the reason for this? How is this an approach that might fit into the realm of personalized medicine?

Author Response

We would like to thank you for taking the time to review our manuscript and for your thoughtful suggestions and insights. We believe that our manuscript has benefited from these insightful suggestions. The detailed comments were all valid. We have carefully considered all reviewer comments, and revised our manuscript as suggested.

Responses to the comments of Reviewer 2

This retrospective study is a well conducted Analysis of a smart technique to perform ERCP on patients with a BillrothII resection. It is well written and undertsandable. Especially the Video illustrates the patterns well. As noted in the limitations, it is a non controlled retrospective observation. Ideally, a control group would give an impression on the relative performance.

Comment #1: Why were patients older than 20 years eligible? How many under 20 received ERCP?

Response #1: We appreciate your suggestion of something that we had not considered. During the study period, there were some patients under the age of 20 who underwent ERCP, but none had a Billroth II anastomosis. We have now deleted “age > 20 years” in the eligibility criteria.

Comment #2: Generally the authors should check for spelling and grammar errors. Also in the provided document the formatization of the document is not well done (different font sizes etc.).

Response #2: As requested, editing for proper grammar, spelling, and expression use was conducted by a native English speaker, and formalization of the document was done.

Comment #3: I do wonder why this very clinical work has been submitted to the Journal of Personalized Medicine. What is the reason for this? How is this an approach that might fit into the realm of personalized medicine?

Response #3: ERCP for Billroth II reconstruction has been a major challenge for pancreatobiliary physicians; therefore, the procedural methodology varies among institutions and outcomes are highly dependent on the operator’s personal skill and perseverance. Hence, in the clinical field, there is an unmet need for a standardized and reproducible ERCP technique that can be applied to most cases of Billroth II reconstruction. By anticipatively obtaining information on the bowel loop formed in the afferent jejunal limb using the road-map technique, we can perform predictable and personalized ERCP procedures tailored to individual anatomical characteristics. We believe that a personalized ERCP procedure based on the analysis of bowel loop patterns in each patient will contribute to improving the level and safety of ERCP procedures in patients with surgically altered anatomy.
